# Low-level factors increase gaze-guidance under cognitive load: A comparison of image-salience and semantic-salience models

**Kerri Walter**⦿*, Peter Bex

Psychology Department, Northeastern University, Boston, MA, United States of America

* walter.ker@northeastern.edu

## Abstract

Growing evidence links eye movements and cognitive functioning, however there is debate concerning what image content is fixated in natural scenes. Competing approaches have argued that low-level/feedforward and high-level/feedback factors contribute to gaze-guidance. We used one low-level model (Graph Based Visual Salience, GBVS) and a novel language-based high-level model (Global Vectors for Word Representation, GloVe) to predict gaze locations in a natural image search task, and we examined how fixated locations during this task vary under increasing levels of cognitive load. Participants (N = 30) freely viewed a series of 100 natural scenes for 10 seconds each. Between scenes, subjects identified a target object from the scene a specified number of trials (N) back among three distracter objects of the same type but from alternate scenes. The N-back was adaptive: N-back increased following two correct trials and decreased following one incorrect trial. Receiver operating characteristic (ROC) analysis of gaze locations showed that as cognitive load increased, there was a significant increase in prediction power for GBVS, but not for GloVe. Similarly, there was no significant difference in the area under the ROC between the minimum and maximum N-back achieved across subjects for GloVe (t(29) = -1.062, p = 0.297), while there was a cohesive upwards trend for GBVS (t(29) = -1.975, p = .058), although not significant. A permutation analysis showed that gaze locations were correlated with GBVS indicating that salient features were more likely to be fixated. However, gaze locations were anti-correlated with GloVe, indicating that objects with low semantic consistency with the scene were more likely to be fixated. These results suggest that fixations are drawn towards salient low-level image features and this bias increases with cognitive load. Additionally, there is a bias towards fixating improbable objects that does not vary under increasing levels of cognitive load.

## Introduction

Fixational eye movements in natural scenes can be driven by both bottom-up, low-level factors and top-down, high-level factors. Low-level factors include basic sensory features, such as contrast, edges, brightness, and color. Several research groups have developed low-level feature-

**Data Availability Statement:** The subject data and images used for this study are available at osf.io/f5dhn.

**Funding:** PB was funded by grant EY032162 from the NIH Research Project Grant Program (NIH R01

EY032162). The funders had no role in study design, data collection and analysis, decision to publish, or preparation of the manuscript.

**Competing interests:** The authors have declared that no competing interests exist.

based models in which the probability of a location being fixated is correlated with image salience [1–4]. Other groups have developed high-level semantic salience based models in which the probability of an image location being fixated is correlated with information-based maps of image and task content [5–11].

### Image salience

While free-viewing a scene, some evidence suggests that gaze [1–4] and overt attention [4] follow a path based on areas that are visually salient–i.e. high local variation in image statistics. A model for visual saliency was originally proposed by Itti, Koch, & Neibur [12] in which local center-surround difference maps of linear filter responses to color, intensity and orientation are combined and weighted to generate local feature salience maps, the peaks of which are sequentially fixated. Many elaborations of this general approach have now been proposed, for recent review see [13, 14]. In this paper, we employ graph-based visual saliency (GBVS), which is one such example that combines three main low-level features: color, edge-orientation, and intensity [2]. We chose the GBVS model due to its robustness and documented ability in predicting human fixations, as well as its accessibility as an open-source toolbox.

### Semantic salience

Contrary evidence suggests that gaze follows a path based on the meaning and context of the setting [5, 15–18] or locations that are relevant for future action [19–21]. These findings support a theory of top-down processes guiding gaze deployment, in which prior experience, knowledge of the world [22], and task objectives [19, 23–25] guide where we look to find relevant information. Note that these changes in knowledge and task are unaccompanied by any change in low-level salience [15, 18, 26]. In this paper, we develop a language-based method utilizing Global Vectors for Word Representation (GloVe) [27] to compute object information based on descriptors of image content [6, 10].

### Cognitive impairment

There is evidence to suggest eye movements and cognitive functioning are interconnected, as individuals with cognitive impairments exhibit abnormal oculomotor behaviors when performing certain tasks. For example, eye movement related correlates have been identified for neurodegenerative diseases such as Alzheimer's, where viewing strategies become more erratic, and saccades and smooth pursuit are slowed [28–31] (for review see [32]). Similarly, children with Autism Spectrum Disorder execute fewer eye movements when processing language and social cues [33, 34], coupled with an exaggerated center-bias when viewing images. It has been proposed that these patterns are driven by reduced attention to gaze locations related to social information processing [34]. In related work, we have demonstrated that oculomotor parameters in neurotypical control subjects are similarly affected by increasing levels of cognitive load. In a demanding visual search task, we observed a decrease in the number of fixations and saccades, coupled with the lengthening of individual fixation durations, under increasing levels of cognitive demand in neurotypical subjects [35].

### Present study

The amount of effort actively invoked by working memory is called cognitive load [36]. Working memory is often measured with N-back tasks as numerous studies have shown that increasing the demands of an N-back task is associated with increased activity in brain regions associated with working memory [37–41]. Based on these findings, we designed an adaptive

N-back protocol that conforms to each subject's individualized working memory load, thus pushing subjects to their discrete maximum cognitive capacity.

Previously, we demonstrated that the manipulation of cognitive load alters the oculomotor characteristics of a participant's search [35]. In the present study, we re-examine the data from the previous paradigm to examine whether cognitive load also alters the low-level and high-level scene content that is fixated. Because top-down processing is affected by cognitive load [42], we hypothesize that the relative contributions of low-level and high-level factors involved in the guidance of gaze will be affected by changing cognitive load. Specifically, we predict that gaze will be guided by semantically relevant objects within the scene (top-down), but as cognitive load increases, gaze will be guided by more visually salient features (bottom-up). Furthermore, we hypothesize that individual differences in working memory capacity (subjects who have high or low performance in the present cognitive load task), will be associated with different viewing strategies. Thus, individuals who are able to identify images from many trials back may utilize more efficient strategies to view and encode images in memory that may be reflected in the information selected. In order to predict gaze guidance, we will use the GBVS model to determine which areas of an image are visually salient and the GloVe model to determine which areas are semantically salient. Because we predict a shift in gaze strategy, we hypothesize that the GloVe model will be a better predictor of gaze under low cognitive load, and GBVS will be a better predictor under high cognitive load.

## Methods

### Apparatus

Our task was programed using MATLAB (The MathWorks, Inc., Natick, MA) with Psychtoolbox [43] and analyzed using MATLAB with the Text Analytics and Statistics and Machine Learning toolboxes. The experiment was run on a Dell OptiPlex 9020 desktop computer (Dell Inc. Round Rock, TX) with a Quadro K420 graphics card (nVidia, Santa Clara, CA). Stimuli were presented on a 60cm x 34cm BenQ XL2720Z LCD monitor (BenQ Corporation, Taipei, Taiwan) set to a screen resolution of 1,920 × 1,080 pixels at 120 Hz. A chinrest was utilized to stabilize the head position of participants, who were seated 63cm from the screen (width = 50.9˚). Eye movements were recorded using an Eyelink 1000 (SR Research Ltd. Mississauga, Ontario, Canada) with the MATLAB Eyelink Toolbox [44], where the sampling rate was set to 1,000 Hz. We used the built-in Eyelink nine-point calibration and validation procedures at the beginning of the experiment and in between blocks.

### Stimuli

We selected 100 images from the LabelMe database [45]; (see [35] for selection criteria), which is a collection of natural scenes in which objects have been outlined and labeled by human volunteers. These scenes and their corresponding labeled annotations are made available for public use and provided us with unbiased, pre-labeled images. 50 indoor and 50 outdoor images were selected from the LabelMe database dependent on each scene containing at least 15 unique objects, having at least 75% of the image labeled, and being a large, clear image. All 100 images were presented in random order to each subject. When presented to participants, all images were resized without cropping or changing the aspect ratio to approximately 1,280 x 960 pixels. An example of an indoor and an outdoor scene used in the experiment can be found in Fig 1.

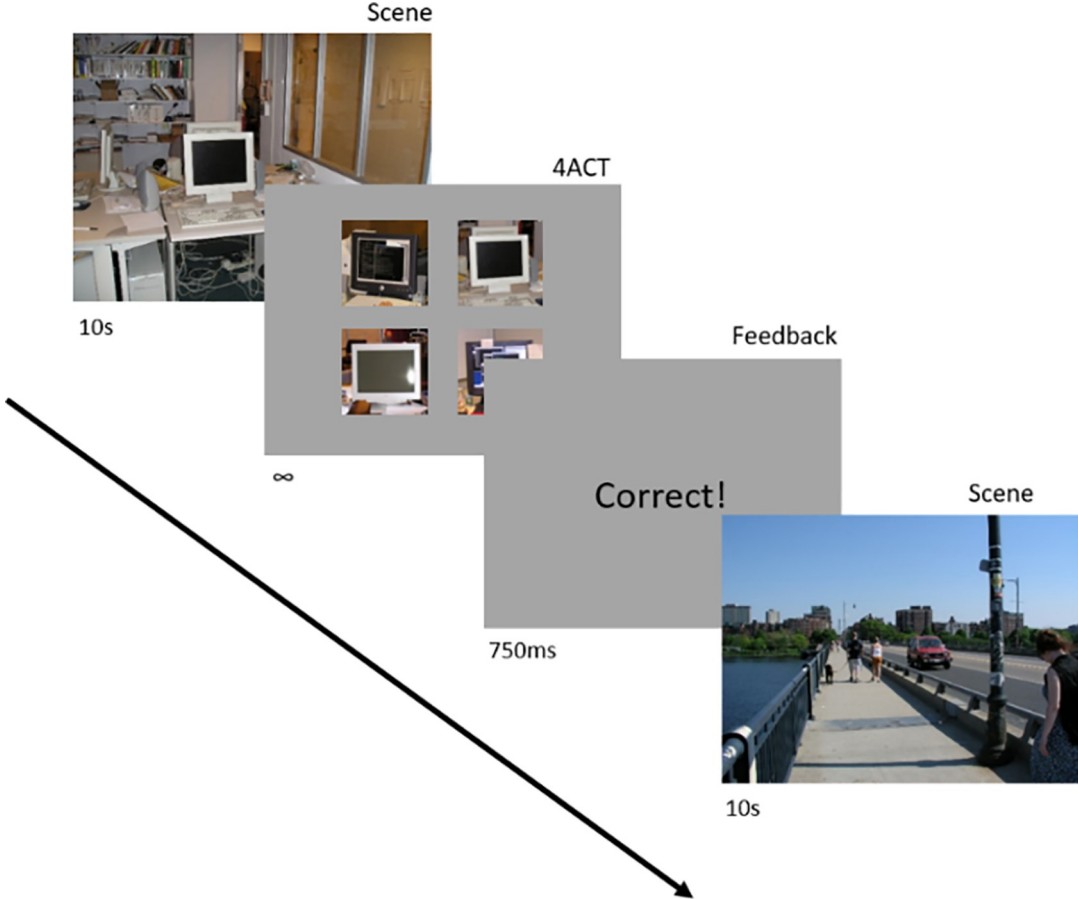

**Fig 1. Illustration of the course of events within the procedure.** Subjects viewed a scene for 10 seconds. The scene was then replaced with a 4AFC Task that was present until the subject responded. The task was to select by mouse click, the object from the image N back in the stream (in this case N = 0, referring to the scene directly preceding the 4AFC Task). For a depiction over a longer course of trials, see [35].

## Participants

We recruited a total of 33 subjects (7 male, 26 female) with self-reported normal or corrected vision from the Northeastern undergraduate population to participate in this study and excluded 3 subjects due to program crashes (N = 2) or Eyelink calibration issues (N = 1), thus we analyzed data for 30 included subjects (7 male, 23 female). We determined a stopping number of 30 before data collection, and tested until 30 useable subjects were obtained. Course credit was given as compensation for each subject's time. Written consent was obtained before the experiment, where all subjects read and signed an informed consent form. Any subject's under the age of 18 received written consent from a parent or guardian. This experiment was performed in accordance with the tenets of the Declaration of Helsinki, and the experimental procedure was approved by the institutional review board at Northeastern University, IRB #: 14-09-16—Psychophysical Study of Visual perception and Eye Movement Control.

## Procedure

As described in [35], participants were shown a scene for 10 seconds and were instructed to view the scene freely. The scene was then replaced with a 4 alternative forced choice task

(4AFC) comprised of similar objects (i.e., having the same word label) where one object was taken from the target scene and three distracter objects were taken from random scenes within the experiment,. For example, if the object were a car, the 4AFC stimulus would contain four cars, where the target car was taken from the target scene, and the other three cars were taken from alternate scenes within the experiment. Objects presented in the 4AFC task were resized to 300 pixels along their longest dimension, while maintaining their original aspect ratio. To prevent overmagnification in the 4AFC display, only objects larger than 100 x 100 pixels were chosen as targets or distracters. Objects were spliced from a rectangular section containing 10% of the surrounding background. This was done to provide minimal context of the object background, because in pilot studies we found that the task was too difficult to complete with no object context. The target object was chosen at random every trial, and the target scene was dependent on the N-back task.

The N-back began at 0, meaning the target object was selected from the scene directly preceding it. Whenever a subject answered two 4AFC trials correctly in a row, they received a prompt informing them that the N-back was increased by 1, with no maximum. The target object would then be selected from the scene 1 back in the image stream. If at any point a subject answered incorrectly, the N-back was decreased by 1, and the subject received a prompt informing them that the N-back was decreased by 1, with a minimum of 0. Subjects received feedback for 750ms, then a new scene was presented, and the cycle continued. Eye movements were recorded at 1000Hz. This adaptive paradigm ensured that all subjects were tested at their individual maximum levels of cognitive load (Fig 1). Lighting conditions were controlled for using blackout curtains, all subjects were tested in the same experimental room using the same equipment, and the luminance of our screen remained constant in an otherwise dark room [46].

## Image salience: Graph Based Visual Saliency (GBVS)

We used the GBVS model developed by Harel and Koch [2], implemented in Matlab (https://www.mathworks.com/products/matlab.html) with the default parameters to compute image salience maps. The GBVS model highlights areas of a scene that have high image salience by using channels of color, orientation, and intensity to create saliency heatmaps. Three individual conspicuity maps are created, coding local variation in each channel, and they are averaged together to create a single salience heatmap [2]. We applied Gaussian smoothing proportional to eye tracking precision, as described below.

## Semantic salience: Global Vectors for Word Representation (GloVe)

For the semantic-based analysis, we highlighted areas of a scene that have high semantic salience by calculating the semantic similarity between a descriptive label for each scene and the label for each object in the scene supplied with each LabelMe image. We used PlacesCNN [47] to generate the scene labels for each of our 100 selected images. PlacesCNN is an algorithm that has been trained on a database of various indoor and outdoor scenes. PlacesCNN analyzes an image based on the content in the scene and provides a scene label of what it assumes the scene location is. For example, a scene with a single desk and computer might be best matched with the scene label "home_office". PlacesCNN returns scene labels in descending order of most-likely scene description, and we selected the top five most-likely scene labels for each of our images. For a list of selected PlacesCNN labels, see S2 Table.

Because the LabelMe database contains noise in the form of junk labels or invalid objects [10], we used the criteria in Table 1 to manually edit object labels before performing our semantic analysis.

**Table 1. List of criteria for manually editing object labels.**

| Criteria | Examples |
|---|---|
| Removed non-words | "gtreve", "aqq", "df", "44" |
| Removed test objects | Areas labeled "test" |
| Fixed spelling errors | "coffe marker" → "coffee maker" |
| Separated conjoined words | "personwalking" → "person walking" |
| Removed unnecessary adjectives | "frontal", "occluded", "crop", "side" |
| Removed obscure shapes | "triangle" over non-discrete area |
| Fixed mislabeled/duplicate label objects | "big brother" over "traffic light" |
| Removed scribble* objects | "sheep pen" |

Criteria and some examples of how the object labels were edited. A full list of these edits can be found in the S1 Table.
*Scribble objects are a type of add-on object in LabelMe. In all cases, these labels were either depicting additional incorrect objects, or were duplicates of properly labeled objects.

We then used the GloVe dataset developed by Stanford University [27] to calculate the similarity for each object in a scene and for each of the top five scene labels. GloVe is a pre-trained regression model that uses both global matrix factorization and local context window methods. GloVe is trained on large web-based datasets, we used the Common Crawl dataset comprised of 840 billion tokens and 2.2 million words. GloVe categorizes words along feature dimensions to create a similarity web of various terms based on multiple dimensions and compare the angles and vector lengths between comparison words to achieve a semantic similarity value between 0 (not similar) and 1 (identical) for any pair of words. For example, the words "office" and "desk" return a semantic similarity value of 0.6319, while "office" and "parrot" return a value of 0.0673. All pixels within the mask area for each object were assigned the semantic similarity value between that object and scene label, and the final semantic salience map was compiled from the average of the individual maps created from the five scene-labels.

The GloVe dataset does not contain dual words, for example there is no "window blinds", but it does contain "window" and "blinds" as separate entities. In order to generate a single word for dual-word objects, we calculated the individual vectors for all components of a multi-word, then used vector math to calculate the closest common word between the multiple vectors. In this example, "window" was the closest relevant word between "window" and "blinds". We performed this step for both objects and scene labels.

An alternative high-level approach entails hand-labeled analysis of image regions to create meaning-maps [5]. This method requires intensive labeling by several human raters, which is impractical for many applications, especially movies. Because the ratings are completed without context or task, this model, like low-level image salience models, is unable to deal with fixation changes with task [18]. By using manually-assigned estimates of meaning, human labelers employ an unknown combination of low-level features and high-level semantic factors that side-steps the problem of these two sources of gaze-guidance and is therefore unsuitable for the present analysis.

## Spatial smoothing

We applied Gaussian smoothing to the heatmaps of both GBVS and GloVe models based on the average manufacturer error reported in the Eyelink 1000 manual. Eyelink reports between 0.25° and 0.50° of error, so we used a standard deviation of 0.375° for the GBVS and GloVe maps. This Gaussian smoothed the transitions between areas of differing significance within the image. We normalized all salience maps from 0–1 before performing an ROC analysis (Fig 2).

**Orignial Image**

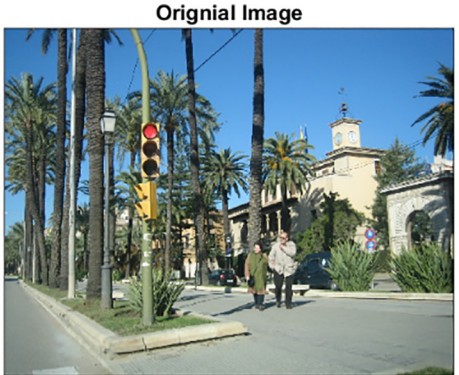

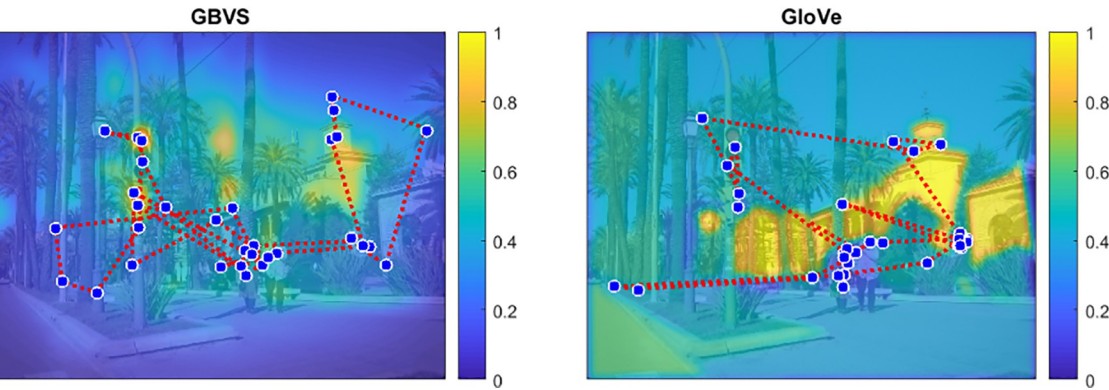

**Fig 2. Examples of one image and salience heatmap overlaid with a representative subject's gaze data.** An example image (a) was analyzed with b) low-level image salience and c) high-level semantic salience, see text for details. Areas with yellow values denote higher relevance coded by each model. A Gaussian of 0.375° was implemented to smooth the heatmap edges between distinct saliencies. Blue circles represent fixations, red dotted lines represent saccades. Heatmap plots demonstrate representative gaze data from 2 individual subjects during the same trial.

## Results

We were successful in actively engaging and increasing working memory load. An increase in working memory load can be measured using response time [38, 39, 41], as well as observing pupil dilation [48], for details, see [35]). As N-back increased, we observed a significant increase in response time compared to N = 0 (t-tests performed comparing N = 0 and all other N-backs, all were significant below .01 except N = 9 and N = 10 due to low sample size), as well as a significant increase when comparing subject's low-load (N = 0) and high-load (N = maximum achieved) conditions (t(29) = −5.717, p < .001). For details, see [35].

For ROC analysis, we set levels of specificity as 100 steps from 0 to 1 (in correspondence with the range of salience values). We quantified true positives as the areas of heatmap within each specificity step that do contain gaze points, true negatives as areas not within the current specificity that do not contain gaze points, false positives as areas within the current specificity that do not contain gaze points, and false negatives as areas not within the current specificity that do contain gaze points. The area under our ROC curves (AUROC) is the prediction power of each map and quantifies how well each heatmap predicted the gaze locations of our subjects [49].

When comparing performance at maximum and minimum load, we performed paired samples t-tests for each salience model. When comparing performance across all Ns, because of the unbalanced nature of our data (N = 0 having 929 cases while N = 10 has 1 case), we performed a linear mixed effect model analysis across N-back. For a more detailed analysis on the distribution of N-back, see [35].

### Permutation analysis

We performed a permutation analysis to determine if any of the models predicted subject's gaze above chance levels. We calculated the AUROC's for corresponding gaze and image for each subject (the subject's gaze pattern taken from the image they viewed, paired with the heatmap for that same image), and compared the AUROC with every other gaze and image non-corresponding combination. To calculate the null distribution, we overlaid gaze data with non-corresponding images, and compared the AUROC values for each map. Because the heatmaps from non-corresponding images are unrelated to the locations of features and objects in the image, the gaze-data should be uncorrelated with the model prediction of gaze locations.

Fig 3 shows the average AUROC value of corresponding gaze and heatmap pairs for each model across all subjects (red stars), plotted along the distribution of AUROC's for all non-corresponding pairs. The GBVS model had a corresponding mean of .731, (z = 0.734; p = .231), and the GloVe model had a corresponding mean of .458, (z = -0.582, p = .280).

The results of this permutation analysis suggest that during the free viewing intervals of the current task, image salience is a better predictor of gaze compared to semantic salience. The observation that the corresponding image and gaze analysis for GLoVe is lower than the midpoint of the null distribution suggests that subjects tend to look at incongruent, or unexpected semantic objects, implying subjects are actively looking at objects that are not relevant to the scene, although this effect was not significant.

### Model summary

Fig 4 shows the overall AUROC for each model, averaged across observers and N-back levels. Overall, the image salience algorithm outperformed the semantic salience model, with a higher average mean AUROC (.731) and lower average standard deviation of AUROC (.079) across subjects. Fixation predictions from the GloVe model did not differ from chance, with a mean AUROC of .458, and had a higher standard deviation of AUROC (.118), demonstrating more

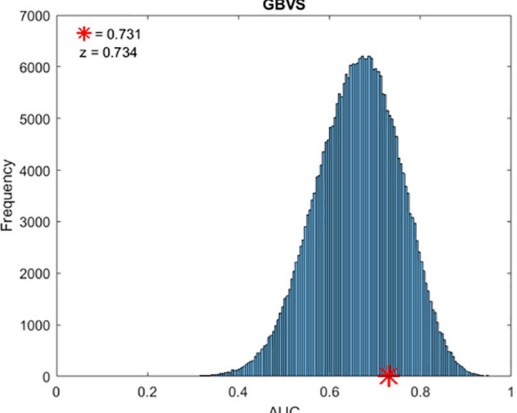 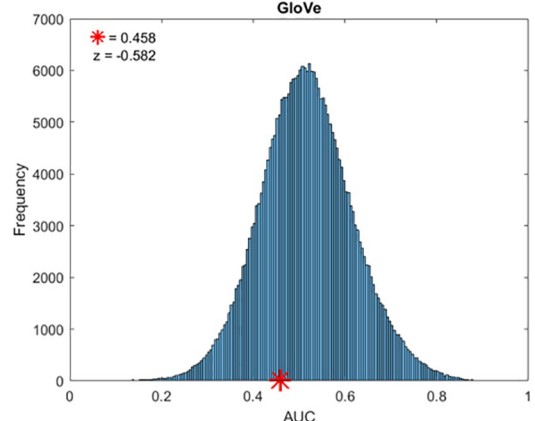

**Fig 3. Permutation analysis for each model.** Red stars represent the AUROC value of the target images and corresponding gaze data averaged across all observers. The histograms represent the permutation set of all combinations of target image and gaze data.

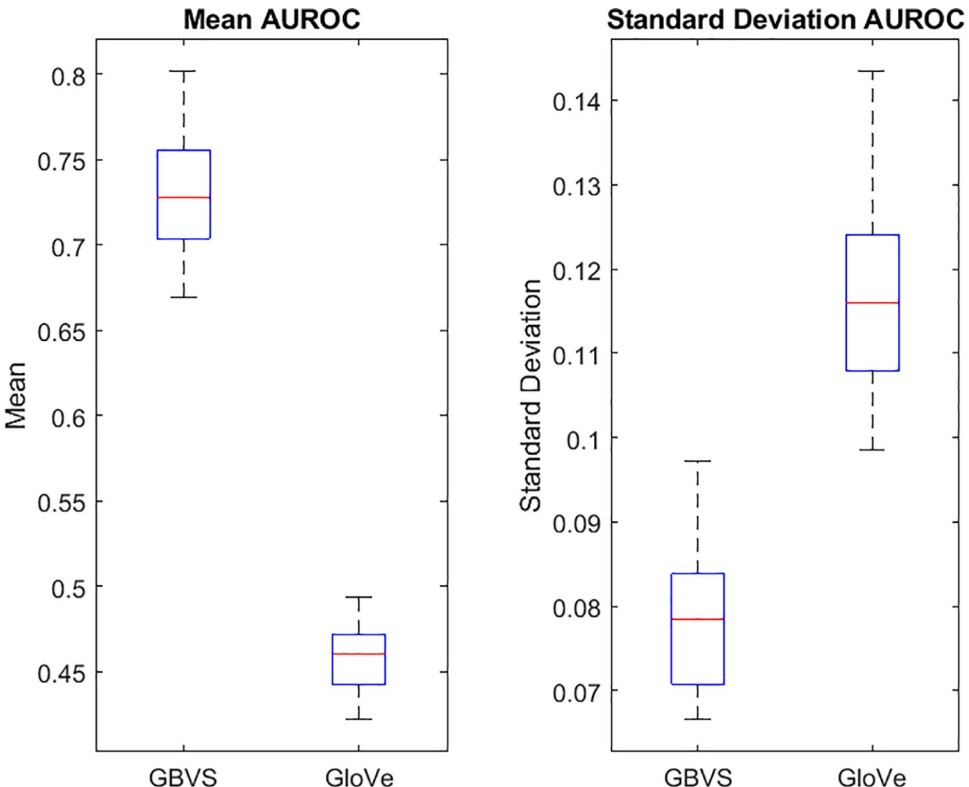

**Fig 4. Mean and standard deviation of AUROC for each model.** AUROC is averaged across all subjects and N-back levels for each subject. Box and whisker plots show the distribution of AUROC for each model, red lines show median, boxes show interquartile range and whiskers show 95% confidence intervals.

variance overall across all subjects. A paired samples t-test shows that these groups are significantly different in both mean (t(29) = 40.662, p < .001) and standard deviation (t(29) = -16.726, p < .001). The results suggest that image salience models provide a better prediction of gaze guidance than the language-based semantic model for this task.

## GBVS

Fig 5 shows the area under the ROC (AUROC) for GBVS as a function of N-back for each subject (faint grey) and as box plots for all subjects. A linear mixed effects model analysis showed a significant effect of N-back for GBVS (X2(1, N = 30) = 8.256, p < .01). This demonstrates that as subjects reach higher N-backs (or as cognitive load increases), gaze is guided more by low-level image salience features. This also suggests that the gaze of subjects who excel at this cognitive load task (those reaching higher N-backs, or those who have high cognitive load capacities), are guided more by low-level image salience features than those who have lower cognitive load capacities.

There were large individual differences in task performance, with subjects reaching a maximum N-back between 2 and 10, with a median of 5 [35]. We therefore used the trials on which each subject reached their individual highest N-back as their maximum load, and N-back = 0 as the lowest load for each subject. Fig 7A shows the image salience model AUROC for minimum and maximum load for each subject (faint grey) and as box plots for all subjects. There was a trend upwards for image salience from minimum to maximum individualized N-back

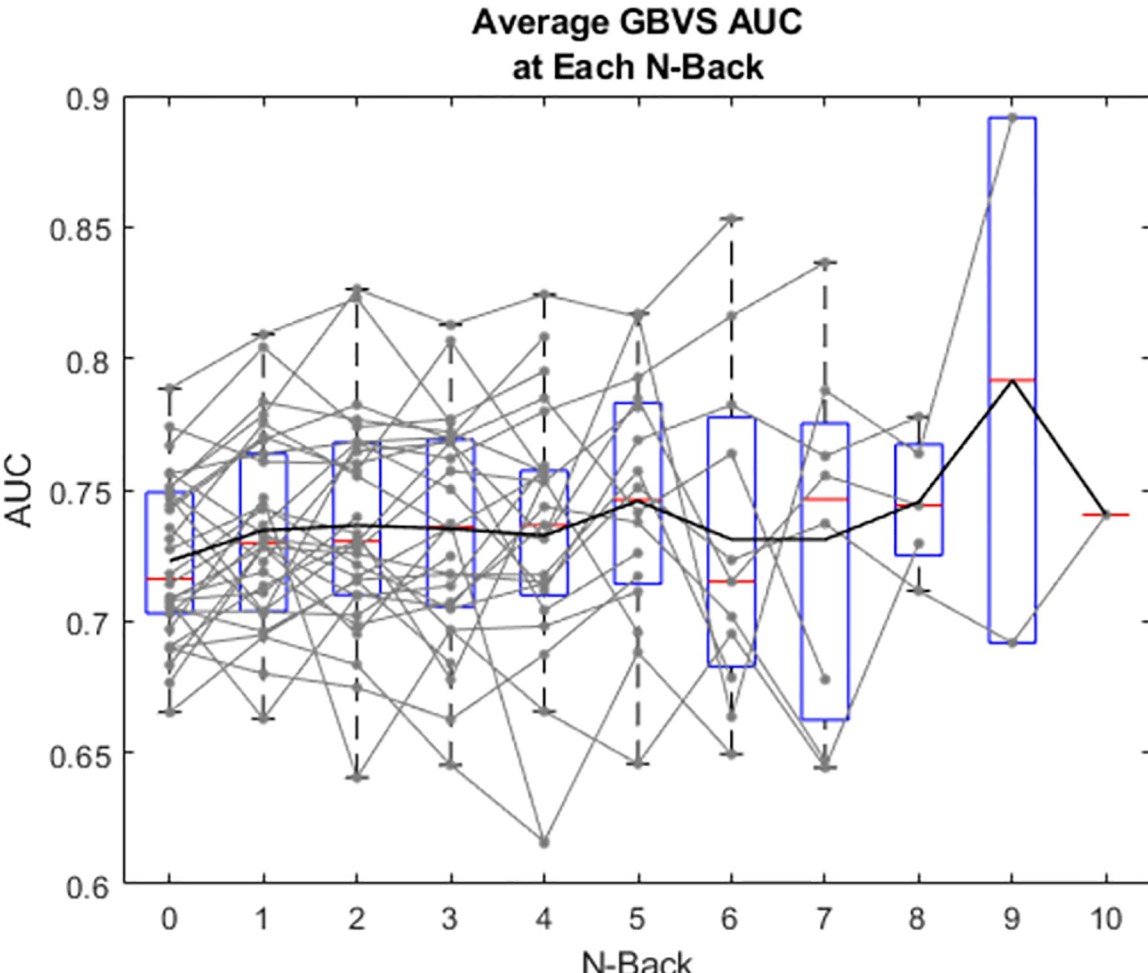

**Fig 5. Area under the ROC curve for GBVS prediction of gaze at each N-back.** Grey lines represent individual subjects. Black line represents mean values, red lines show median, boxes show interquartile range and whiskers show 95% confidence intervals.

across subjects, however this did not reach significance (t(29) = -1.975, p = .058). This is consistent with a trend in which gaze becomes more biased towards salient low-level image features as cognitive load increased in the present task.

## GloVe

Fig 6 shows the AUROC as a function of N-back for GloVe for each subject (faint grey) and as box plots for all subjects. A linear mixed effects analysis showed no effect of N-back for GloVe ($X2(1, N = 30) = 1.487$, p = .223). This suggests that the gaze of subjects who excel at this cognitive load is not biased towards image locations with higher or lower semantic similarity in the scene compared to those who have low cognitive load capacities.

Fig 7B shows the GloVe semantic model AUROC for minimum and maximum load for each subject (faint grey) and as box plots for all subjects. There was no significant difference between AUROC for the minimum and maximum N-back achieved across subjects (t(29) = -1.062, p = .297). This suggests that the semantic image-viewing strategy was unaffected by cognitive load in the present task.

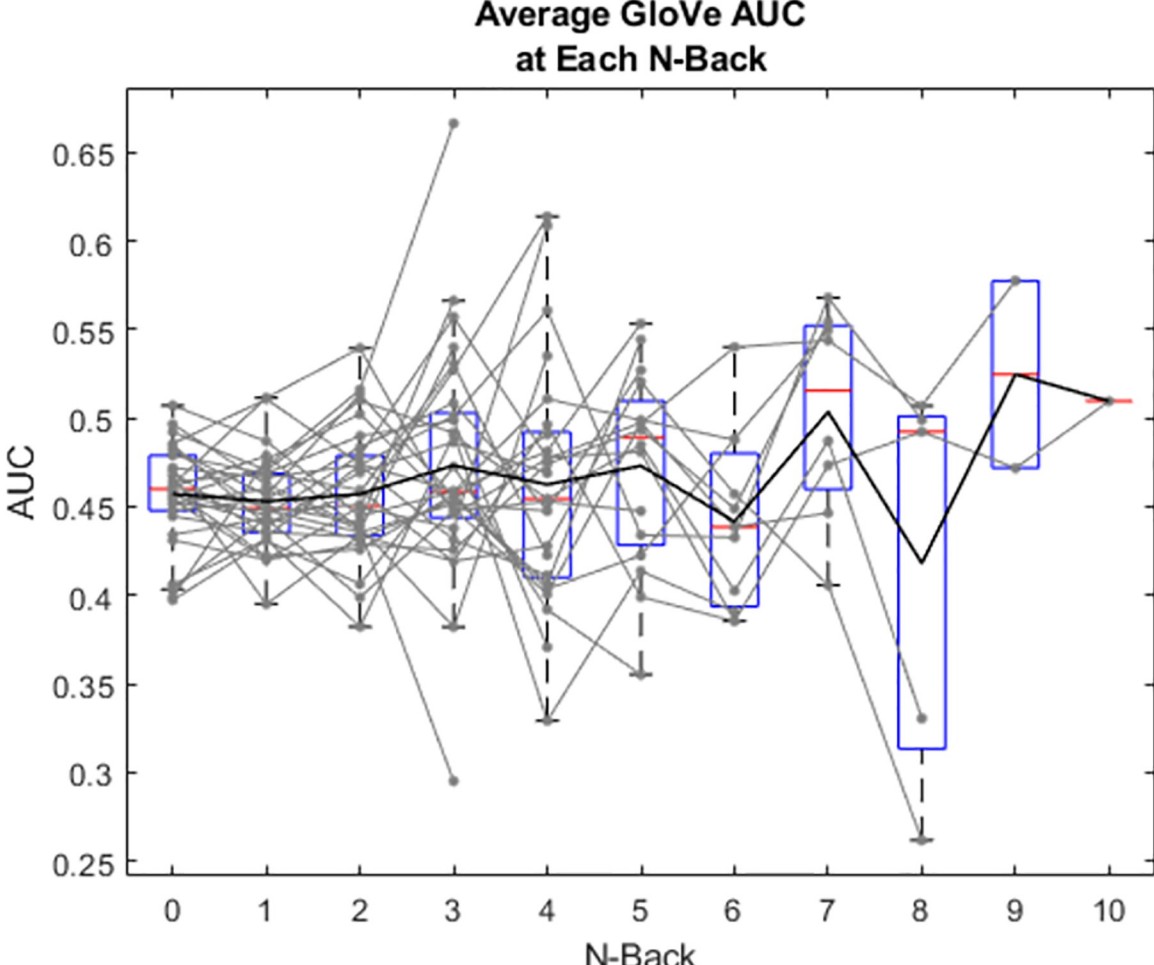

**Fig 6. Area under the ROC curve for GloVe-based predictions of gaze at each N-back.** As Fig 5.

## Discussion

We move our eye 2–3 times per second in order to position our high-resolution fovea on image locations of interest. This process involves oculomotor processes that coordinate these binocular eye movements, sensory processes that encode image features and cognitive processes that estimate the locations of task-relevant objects. Growing evidence in neurological studies has identified an association between eye movement behavior and cognitive processing and we recently identified a change in oculometrics as a function of cognitive load in healthy young subjects. Specifically, with increasing cognitive load we observed a decrease in number of fixations and saccades, and an increase in fixation duration [35], an effect that is similar to differences observed between neurotypical and cognitively impaired populations (for review see [32]). In this study, we hypothesized that cognitive load could be associated with changes in low-level and high-level factors involved in viewing strategy.

Due to the nature of our task, which was task independent at the time of scene viewing, subjects did not know which object would become the target when they viewed each scene. This created an implied information-gathering task for scene viewing. Therefore, we assumed that subjects would attempt to generate a broad mental representation of the full scene during the viewing phase, while maintaining a memory of previous scenes for the identification tasks. We

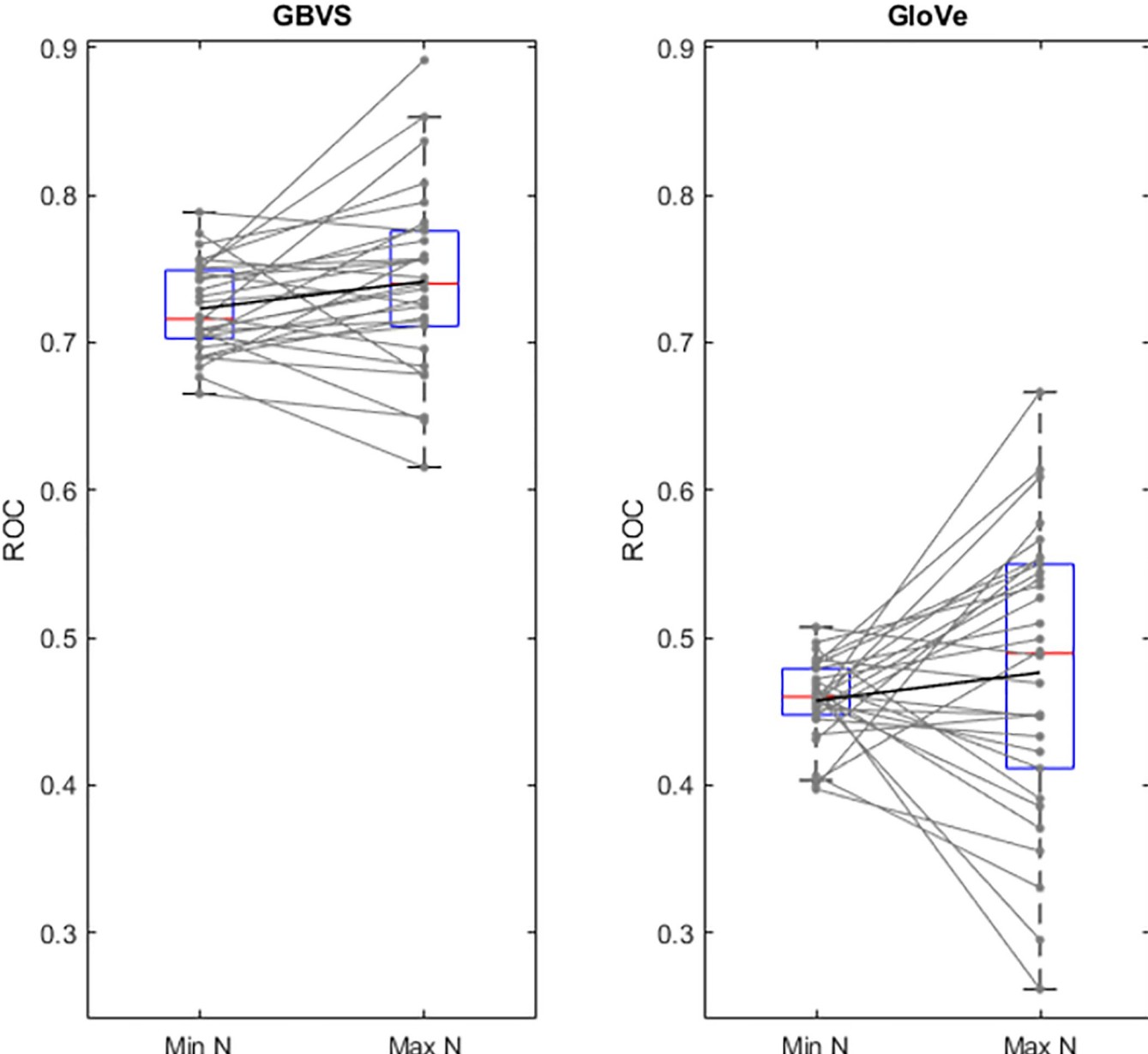

**Fig 7. Area under the ROC at minimum and maximum cognitive load for each model.** Grey lines represent individual traces. Black line represents mean values. Minimum N (minimum load) for all subjects was N = 0, maximum N (maximum load) was variable across subjects. Red lines show median, boxes show interquartile range and whiskers show 95% confidence intervals.

hypothesized that as cognitive load increased, gaze guidance would shift towards low-level sensory factors and away from high-level cognitive factors. We quantified sensory gaze guidance with an image salience model, and cognitive gaze-guidance with a language-based semantic models.

Overall, the AUROC for image salience was greater than for semantic salience, indicating that low-level features generally predict fixation locations in natural scenes for the present task better than high-level semantic content. The prediction power of the image salience model (GBVS), significantly increased with increasing cognitive load. This suggests that as cognitive

load increased, the role of low-level image features in directing fixations increased. Contrarily, the prediction power of the semantic model (GloVe), did not significantly change with cognitive load. The increase in prediction power of the GBVS model across N-back suggests that subjects who excel at this task, or those who were able to reach higher N-backs, utilize a more image salience based viewing strategy compared to those who struggle with this task, or those who could only reach lower N-backs.

In addition to comparing AUROC as a function of N-back, we also examined AUROC at the minimum and maximum N-back achieved on an individual subject basis. This comparison was not significant for either GBVS or GloVe models (albeit, was close to reaching significance for GBVS). However, Fig 7 demonstrates more consistency in the GBVS model, and more consistency in the individual subject trends. Although this overall trend did not reach significance, it demonstrates a more coherent tendency of an image salience based viewing strategy under increased cognitive load.

An additional explanation could be that task relevance is entirely responsible for shifts in related gaze. It has already been demonstrated that task influences gaze [19, 23–25]. In our paradigm, the task required subjects to utilize memory in order to recall objects within scenes. Because this task is based in semantics, we saw a decrease in gaze within visually salient areas with increased cognitive load. However, how would we expect gaze to shift in a task where an increase in cognitive load requires searching for more visually salient features? For example, walking through a cluttered area at increasing speed–where an increase in cognitive load (speed at which you must scan the environment and avoid obstacles) is reliant on attention to visually salient features. In this case, perhaps the opposite result would be true: an increase in load would result in a decrease in gaze towards semantically salient features. As we only have empirical evidence for a semantically related task we cannot draw conclusions towards a visually salient one, however further research into this area may hold interesting findings on how task influences gaze.

Although there was no significant relationship between AUROC and cognitive load for GloVe model of semantic salience, our permutation analysis demonstrated that the GloVe model predicted gaze less than chance, meaning there was a negative correlation between semantic salience and the probability of an object being fixated. This finding indicates that observers may be more likely to fixate an 'outlier' object that is inconsistent with its semantic context than an object that is semantically related to its context.

A potential confounding factor in our design is the frequency of certain objects in our images. For example, windows occur in both outdoor and indoor scenes, and in the case of large buildings, there are numerous windows in one image. This led to "window" being a common search term due to the random nature of our object selection process, and as a result "window" was chosen more often than other objects. Observant participants may pick up on this trend and change their search strategy to scan for windows. Future iterations of this task should include a failsafe to ensure common objects such as "window" are not chosen as a search term more often than other terms. A solution would be to condition the random selection of search objects upon their frequency, so that the resulting sampling be uniform per unique item.

The GBVS model was a better predictor overall with this task. This suggests that image salience is the driving factor when searching during our task, rather than semantics as quantified by our language-based model. The viewing portion of our paradigm did not have an explicit task, although wide exploration of the scene is explicit given that the search item in an upcoming trial was unknown, this finding provides evidence that image salience guides eye-movements during exploration of natural scenes. It is possible that semantic models may have greater predictive power when observers are assigned an explicit task, such as to search for a

specific object (e.g. window), in which case we would expect observers to look at window-like objects. Additionally, the GBVS model predicted human fixations increasingly greater with increased cognitive load. This implies that as cognitive load increases, gaze guidance relies more on sensory factors (image salience) and less on cognitive factors (semantic salience).

## Supporting information

**S1 Table. List of all manually edited object labels from the LabelMe database.** (XLSX)

**S2 Table. Top 5 scene labels obtained from PlacesCNN for each image.** (XLS)

## Author Contributions

**Conceptualization:** Kerri Walter, Peter Bex.

**Data curation:** Kerri Walter.

**Formal analysis:** Kerri Walter.

**Funding acquisition:** Peter Bex.

**Supervision:** Peter Bex.

**Writing – original draft:** Kerri Walter.

**Writing – review & editing:** Kerri Walter, Peter Bex.

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
