## [Decision Letter · Decision Letter 0]

9 Aug 2022

PONE-D-22-06433Low-Level Factors Increase Gaze-Guidance Under Cognitive Load: A  Comparison of Image-Salience and Semantic-Salience ModelsPLOS ONE

Dear Dr. Walter,

Thank you for submitting your manuscript to PLOS ONE. After careful consideration, we feel that it has merit but does not fully meet PLOS ONE’s publication criteria as it currently stands. Therefore, we invite you to submit a revised version of the manuscript that addresses the points raised during the review process. Can you please address the expert reviewer's comments thoroughly?

We look forward to receiving your revised manuscript.

Kind regards,

Avanti Dey, PhD

Staff Editor

PLOS ONE

https://journals.plos.org/plosone/s/file?id=ba62/PLOSOne_formatting_sample_title_authors_affiliations.pdf".

“Supported by NIH R01 EY029713.”

“PB was funded by grant EY029713 from the NIH Research Project Grant Program (https://www.nih.gov/). The funders had no role in study design, data collection and analysis, decision to publish, or preparation of the manuscript.”

5. Please amend either the title on the online submission form (via Edit Submission) or the title in the manuscript so that they are identical.

6. Please upload a copy of Figure 8, to which you refer in your text on page 15. If the figure is no longer to be included as part of the submission please remove all reference to it within the text.

Reviewers' comments:

Reviewer's Responses to Questions

**Comments to the Author**

1. Is the manuscript technically sound, and do the data support the conclusions?

Reviewer #1: Partly

2. Has the statistical analysis been performed appropriately and rigorously? 

Reviewer #1: Yes

3. Have the authors made all data underlying the findings in their manuscript fully available?

Reviewer #1: Yes

4. Is the manuscript presented in an intelligible fashion and written in standard English?

Reviewer #1: Yes

5. Review Comments to the Author

Reviewer #1: Summary

This manuscript reports the results of one experiment that was aimed at testing how the contributions of low-level and high-level processes to gaze guidance change with cognitive load. The main hypothesis is that low-level factors should contribute more as cognitive load increases, much like individuals with cognitive impairment exhibited abnormal oculomotor behavior under certain conditions. Subjects viewed a series of images of natural scenes for 10 seconds each with no gaze restrictions (free viewing). Between images, they were presented with four objects and had to choose the object from the image N trials back. The N-back was adaptive in that N increased following two correct responses and decreased following one incorrect response. The authors used two models to generate predictions about fixation locations for each image: a low-level, saliency-based model (GBVS) and a high-level, language-based model (GloVe). The GBVS model outperformed the GloVe model in that it generated predictions of fixations that were both more accurate and (apparently) varied with cognitive load. The authors conclude that low-level saliency is the best predictor of fixations and that the bias to look toward salient low-level image features increases with cognitive load.

Overall evaluation

My overall evaluation of this paper is mixed. On the one hand, the study explores an interesting hypothesis that has both theoretical and practical significance. Their approach makes clever use of the N-back task, and they employed established models to generate competing predictions. The manuscript was generally well-written and easy to follow. On the other hand, there are several issues that I feel need to be addressed before the manuscript could be considered for publication.

Major/general issues

I felt that the logic of the approach was not explained as clearly as it could have been. Specifically, it was not immediately clear to me why it was necessary or useful to calculate the similarity between each object in an image and the scene label (as described on p. 9). If I understand correctly, the assumption is that if gaze is guided by top-down processes, then observers will fixate objects that are similar to overall description of a scene (e.g., if the image depicts an office, they should look at the computer or the desk). This is a key part of the logic of the experiment but was never explicitly stated. I would recommend adding a few sentences to the end of the introduction and then restating it on p. 9 to make this point clear.

The main conclusion rests on the (apparently) significant effect of N-back in Figure 5, interpreting a borderline non-significant effect of N-back in Figure 7A as meaningful, and interpreting the non-significant effect in Figures 6 and 7B as no effect of N-back. Overall, the evidence seems rather weak. Looking at the figures, it is hard to see an effect in Figure 5 and 7A but not in Figure 6 and 7B. Can the authors explain this? Wouldn’t it make more sense to base the interpretation on the interaction between N-back and model?

How broadly would the authors like to generalize their conclusion that “as cognitive load increases, gaze guidance relies more on sensory factors (image salience) and less on cognitive factors (semantic salience)” (last sentence of the manuscript, middle of p. 16). I have two reasons for asking. First, the GLoVe model is not the only way to quantify semantic salience. There could be other approaches. For example, perhaps observers look at objects that reduce their uncertainty about the overall meaning of the scene. A model based on this assumption would generate a different set of semantic saliency maps that might better fit the human data. Second, I could also see the effect of cognitive load working in the other direction in some circumstances. For example, when walking at a leisurely pace on a relatively flat hiking trail (low cognitive load), gaze may be guided by low-level image salience (e.g., the motion of an animal scurrying in the woods, a bright yellow flower). If the terrain becomes more challenging and the hiker starts running (high cognitive load), I would think that gaze would be driven less by image salience and more by task-relevance (i.e., a top-down process). Wouldn’t this example suggest the opposite of the main conclusion of this study?

Minor/specific issues

p. 3: The authors explained that they chose to use the GBVS model to generate predictions based on the low-level saliency account. What was the justification for choosing this model rather than one of the other saliency-based models? Would the results have turned out any different if one of the alternative models was used?

p. 3 (bottom): If you’re looking to cite papers that provide evidence of the effects of task on gaze behavior during walking, I would recommend the recent paper by Dominguez-Zamora & Marigold (2021) in Current Biology.

p. 5 (top): Please clarify the differences between this study and Walter & Bex (2021, Sci Rep). The two seem very similar but there is only a single sentence (top of p. 5) that speaks to the difference.

p. 6: At this point, it would be useful to the reader to see some examples of images used in the experiment.

p. 6: What was the justification for the decision to use 33 subjects?

p. 8 (middle): I found the first paragraph of the section on semantic salience to be a bit confusing. The authors refer to a “descriptive label for each scene” and a process for calculating the semantic similarity between the scene label and object label, as if these terms/concepts and their meaning in the context of this study were already introduced. They are eventually unpacked on the next page but the authors might consider letting readers know that so that they understand that further explanation is forthcoming.

p. 9 (middle): It would help to be provided with an example or two object-scene pairings that received a high and low ratings.

p. 9 (bottom): If I understand correctly, the alternative approach from Henderson et al. (2019) is an alternative to generating a map of semantic salience. However, because it appears in the middle of a paragraph about how the authors dealt with the problem of dual-word objects, I initially assumed that it was an alternative approach to dealing with that problem.

p. 10 (bottom): The authors report that as N-back increase, they “observed a significant increase in response time compared to N=0”. However, they did not report a statistical test result or show this result in a figure. (The t-test at the end of the paragraph refers to a different comparison.)

p. 11 (top): Please clarify what it means to “set levels of specificity as 100 steps from 0 to 1”. Specificity of what?

p. 13 (top): The authors’ interpretation of the results depicted in Figure 5 was that “the gaze of subjects who excel at this cognitive load task (those reaching higher N-backs, or those who have high cognitive load capacities), are guided more by low-level image salience features than those who have lower cognitive load capacities.” While this may be correct, I’m not entirely convinced that this is what is shown by Figure 5. I think what they mean is that when subjects reached higher N-backs, their gaze was guided more by low-level image salience features than when they were at lower N-backs. This is not the same as what is written at the top of p. 13.

p. 15 (middle): There is a reference to Figure 8 but no Figure 8 in the manuscript.

6. PLOS authors have the option to publish the peer review history of their article (what does this mean?). If published, this will include your full peer review and any attached files.

Reviewer #1: No

---

## [Author Response · Author response to Decision Letter 0]

12 Sep 2022

https://journals.plos.org/plosone/s/file?id=ba62/PLOSOne_formatting_sample_title_authors_affiliations.pdf". 

The manuscript and corresponding files have been edited to reflect the style requirements of PLOS ONE. 

Additional information concerning participant consent has been added to the Methods section and online submission form. 

We did not have access to the ‘Funding Information’ or ‘Financial Disclosure’ sections when attempting to resubmit with revisions on the editorial manager portal – the correct grant number is NIH R01 EY029713. 

“Supported by NIH R01 EY029713.” 

“PB was funded by grant EY029713 from the NIH Research Project Grant Program (https://www.nih.gov/). The funders had no role in study design, data collection and analysis, decision to publish, or preparation of the manuscript.” 

We approve this funding statement. 

5. Please amend either the title on the online submission form (via Edit Submission) or the title in the manuscript so that they are identical. 

6. Please upload a copy of Figure 8, to which you refer in your text on page 15. If the figure is no longer to be included as part of the submission please remove all reference to it within the text. 

The figure referenced on page 15 has been corrected to Figure 7. 

Reviewer's Responses to Questions 

Comments to the Author 

1. Is the manuscript technically sound, and do the data support the conclusions? 

Reviewer #1: Partly 

2. Has the statistical analysis been performed appropriately and rigorously? 

Reviewer #1: Yes 

3. Have the authors made all data underlying the findings in their manuscript fully available? 

Reviewer #1: Yes 

4. Is the manuscript presented in an intelligible fashion and written in standard English? 

Reviewer #1: Yes 

5. Review Comments to the Author 

Reviewer #1: Summary 

This manuscript reports the results of one experiment that was aimed at testing how the contributions of low-level and high-level processes to gaze guidance change with cognitive load. The main hypothesis is that low-level factors should contribute more as cognitive load increases, much like individuals with cognitive impairment exhibited abnormal oculomotor behavior under certain conditions. Subjects viewed a series of images of natural scenes for 10 seconds each with no gaze restrictions (free viewing). Between images, they were presented with four objects and had to choose the object from the image N trials back. The N-back was adaptive in that N increased following two correct responses and decreased following one incorrect response. The authors used two models to generate predictions about fixation locations for each image: a low-level, saliency-based model (GBVS) and a high-level, language-based model (GloVe). The GBVS model outperformed the GloVe model in that it generated predictions of fixations that were both more accurate and (apparently) varied with cognitive load. The authors conclude that low-level saliency is the best predictor of fixations and that the bias to look toward salient low-level image features increases with cognitive load. 

Overall evaluation 

My overall evaluation of this paper is mixed. On the one hand, the study explores an interesting hypothesis that has both theoretical and practical significance. Their approach makes clever use of the N-back task, and they employed established models to generate competing predictions. The manuscript was generally well-written and easy to follow. On the other hand, there are several issues that I feel need to be addressed before the manuscript could be considered for publication. 

Major/general issues 

I felt that the logic of the approach was not explained as clearly as it could have been. Specifically, it was not immediately clear to me why it was necessary or useful to calculate the similarity between each object in an image and the scene label (as described on p. 9). If I understand correctly, the assumption is that if gaze is guided by top-down processes, then observers will fixate objects that are similar to overall description of a scene (e.g., if the image depicts an office, they should look at the computer or the desk). This is a key part of the logic of the experiment but was never explicitly stated. I would recommend adding a few sentences to the end of the introduction and then restating it on p. 9 to make this point clear. 

We have added some clarification on the logic of the experiment at the end of the 

“Present study” section of the introduction and reiterated it within the “Semantic salience” section of the procedure (we have also rearranged this section as per a later comment). 

The main conclusion rests on the (apparently) significant effect of N-back in Figure 5, interpreting a borderline non-significant effect of N-back in Figure 7A as meaningful, and interpreting the non-significant effect in Figures 6 and 7B as no effect of N-back. Overall, the evidence seems rather weak. Looking at the figures, it is hard to see an effect in Figure 5 and 7A but not in Figure 6 and 7B. Can the authors explain this? Wouldn’t it make more sense to base the interpretation on the interaction between N-back and model? 

Although the figures are similar, the results of our mixed linear model found a significant increase for GBVS and no significant difference for GloVe. We believe this is due to an overall more consistent agreement of trends for GBVS compared to GloVe. This can be seen more clearly in Fig7, where the individual subject lines more consistently trend upwards in A compared to B. 

How broadly would the authors like to generalize their conclusion that “as cognitive load increases, gaze guidance relies more on sensory factors (image salience) and less on cognitive factors (semantic salience)” (last sentence of the manuscript, middle of p. 16). I have two reasons for asking. First, the GLoVe model is not the only way to quantify semantic salience. There could be other approaches. For example, perhaps observers look at objects that reduce their uncertainty about the overall meaning of the scene. A model based on this assumption would generate a different set of semantic saliency maps that might better fit the human data. Second, I could also see the effect of cognitive load working in the other direction in some circumstances. For example, when walking at a leisurely pace on a relatively flat hiking trail (low cognitive load), gaze may be guided by low-level image salience (e.g., the motion of an animal scurrying in the woods, a bright yellow flower). If the terrain becomes more challenging and the hiker starts running (high cognitive load), I would think that gaze would be driven less by image salience and more by task-relevance (i.e., a top-down process). Wouldn’t this example suggest the opposite of the main conclusion of this study? 

This is a great point which allowed us to think a bit deeper about our results. We believe our results are indicative of the subject attending to the primary task at hand, in this case remembering objects. Here we show that specifically with memory, subjects gaze locations were less well-predicted by image salience with increased load (as the task, remembering objects, was semantic based). Because we only have empirical evidence for this task, it would be interesting to have evidence for other tasks, e.g. the hiking example mentioned here. If these results are strictly task relevant, then we predict that observers would fixate task-relevant locations (e.g. secure footholds that may not be correlated with image salience). Although it is a matter for empirical determination, we agree with the reviewer that we would expect to see the opposite effect of cognitive load on fixations for a secondary task in the hiking example, where task-relevant areas would be more image salience based, and as such an increase in load would result in a decrease of gaze in semantic salience related areas. We have included some discussion about this in the discussion section. 

Minor/specific issues 

p. 3: The authors explained that they chose to use the GBVS model to generate predictions based on the low-level saliency account. What was the justification for choosing this model rather than one of the other saliency-based models? Would the results have turned out any different if one of the alternative models was used? 

We chose the GBVS model due to its availability as an open-source toolbox and its robustness and evaluated success in predicting human fixations. Because the majority of salience models utilize most of the same main filtering channels (ie. color, contrast, intensity), we believe our results would be largely unchanged if a different model were used. We have added this justification for our choice of model in the text. 

p. 3 (bottom): If you’re looking to cite papers that provide evidence of the effects of task on gaze behavior during walking, I would recommend the recent paper by Dominguez-Zamora & Marigold (2021) in Current Biology. 

Thank you for the relevant paper, we have added it as a reference on how task objective impacts gaze guidance. 

p. 5 (top): Please clarify the differences between this study and Walter & Bex (2021, Sci Rep). The two seem very similar but there is only a single sentence (top of p. 5) that speaks to the difference. 

The study is the same as that performed in Walter & Bex 2021, analyzed differently. The eye movement data obtained in Walter & Bex 2021 are the same data used here, however the previous study analyzed the oculomotor metrics while this study analyzes the salience metrics at fixated locations. This has been clarified in the text. 

p. 6: At this point, it would be useful to the reader to see some examples of images used in the experiment. 

We have directed attention to Fig 1 within the stimuli paragraph, as it contains some examples of images used in the experiment. We did not move this Fig up in the text because it demonstrates the procedure and would be confusing to show before the procedure is explained. However, making note of the Fig within the stimuli section should direct readers to it for examples of images used in the experiment. 

p. 6: What was the justification for the decision to use 33 subjects? 

This was an exploratory investigation with novel endpoints and therefore we could not conduct a power analysis to estimate sample size for a target effect size. As a conservative approach, we determined a stopping number of 30 before data collection and tested until 30 usable subjects were obtained (the three subjects that were excluded were known to be excluded immediately after their participation). This has been clarified in the text. 

p. 8 (middle): I found the first paragraph of the section on semantic salience to be a bit confusing. The authors refer to a “descriptive label for each scene” and a process for calculating the semantic similarity between the scene label and object label, as if these terms/concepts and their meaning in the context of this study were already introduced. They are eventually unpacked on the next page but the authors might consider letting readers know that so that they understand that further explanation is forthcoming. 

We have moved the explanation of how descriptive scene labels were obtained to before Table 1, within the same paragraph they are introduced. 

p. 9 (middle): It would help to be provided with an example or two object-scene pairings that received a high and low ratings. 

We have included an example of a high and low pairing; “office” and “desk” (0.6319) compared to “office” and “parrot” (0.0673). 

p. 9 (bottom): If I understand correctly, the alternative approach from Henderson et al. (2019) is an alternative to generating a map of semantic salience. However, because it appears in the middle of a paragraph about how the authors dealt with the problem of dual-word objects, I initially assumed that it was an alternative approach to dealing with that problem. 

We have separated this section into its own paragraph to clarify that these are individual ideas. 

p. 10 (bottom): The authors report that as N-back increase, they “observed a significant increase in response time compared to N=0”. However, they did not report a statistical test result or show this result in a figure. (The t-test at the end of the paragraph refers to a different comparison.) 

Response latency results are reported in more depth in Walter & Bex, 2021, we have added a citation in case readers are interested in exploring those results further. We did not want to re-use figures in an alternate journal for copyright reasons, however felt a note about response time was important in both papers. Due to the unbalanced sample sizes for each of our N-back groups, we ran conservative t-tests for each N-back, and for simplicity, plotted the significant results against N=0. All N-backs were significant at or below .01 except for N=9 and N=10, due to their low sample sizes. We have made a statistical note in the text and have referred readers to the earlier paper. 

p. 11 (top): Please clarify what it means to “set levels of specificity as 100 steps from 0 to 1”. Specificity of what? 

The range of specificity levels corresponds to the range of salience values. In order to perform the ROC analysis, we must test the salience value of each pixel across the image with increasing specificity. For example, if the first level of specificity is all salience values between 0 and .01, the next is all salience values between .01 and .02, etc. We have included a note in the text that the specificity is in correspondence with the range of salience values. 

p. 13 (top): The authors’ interpretation of the results depicted in Figure 5 was that “the gaze of subjects who excel at this cognitive load task (those reaching higher N-backs, or those who have high cognitive load capacities), are guided more by low-level image salience features than those who have lower cognitive load capacities.” While this may be correct, I’m not entirely convinced that this is what is shown by Figure 5. I think what they mean is that when subjects reached higher N-backs, their gaze was guided more by low-level image salience features than when they were at lower N-backs. This is not the same as what is written at the top of p. 13. 

We have added a sentence to relate directly back to Figure 5 that explains increased cognitive load results in gaze-guidance through low-level features, and kept our original statement as an additional conclusion, less so directly related to Figure 5 itself. 

p. 15 (middle): There is a reference to Figure 8 but no Figure 8 in the manuscript. 

The figure referenced on page 15 was corrected to Figure 7.

---

## [Decision Letter · Decision Letter 1]

5 Oct 2022

PONE-D-22-06433R1Low-level factors increase gaze-guidance under cognitive load: a comparison of image-salience and semantic-salience modelsPLOS ONE

Dear Dr. Walter,

Thank you for submitting your manuscript to PLOS ONE. After careful consideration, we feel that it has merit but does not fully meet PLOS ONE’s publication criteria as it currently stands. Therefore, we invite you to submit a revised version of the manuscript that addresses the points raised during the review process.

We look forward to receiving your revised manuscript.

Kind regards,

Marcela de Lourdes Peña Garay, Ph.D

Academic Editor

PLOS ONE

Journal Requirements:

Additional Editor Comments (if provided):

Please, we need that you consider to rephrase the comment from reviewer 1 on page 6.

Reviewers' comments:

Reviewer's Responses to Questions

**Comments to the Author**

1. If the authors have adequately addressed your comments raised in a previous round of review and you feel that this manuscript is now acceptable for publication, you may indicate that here to bypass the “Comments to the Author” section, enter your conflict of interest statement in the “Confidential to Editor” section, and submit your "Accept" recommendation.

Reviewer #1: (No Response)

2. Is the manuscript technically sound, and do the data support the conclusions?

Reviewer #1: Yes

3. Has the statistical analysis been performed appropriately and rigorously? 

Reviewer #1: Yes

4. Have the authors made all data underlying the findings in their manuscript fully available?

Reviewer #1: Yes

5. Is the manuscript presented in an intelligible fashion and written in standard English?

Reviewer #1: Yes

6. Review Comments to the Author

Reviewer #1: The authors have addressed all of my comments with one exception. I feel that their characterization of the relation between the present study and Walter & Bex (2021) was a bit vague. On p. 6, they wrote that "In the present study, we utilize the same paradigm...". If I understand correctly, they used the same data. In other words, in the present study, the authors reanalyzed the data that was published in the previous study. The new analyses are sufficiently different from those of the previous study, so I'm not suggesting that this is a problem. My point is simply that it should be clear to the reader that the data are the same.

Other than this one minor point, I think that the paper is suitable for publication. The authors did an excellent job responding to my concerns and the revised version will make an interesting contribution to the literature.

7. PLOS authors have the option to publish the peer review history of their article (what does this mean?). If published, this will include your full peer review and any attached files.

Reviewer #1: No

---

## [Author Response · Author response to Decision Letter 1]

7 Oct 2022

Journal Requirements: 

The manuscript has been checked and all references are current and relevant. 

Additional Editor Comments (if provided): 

Please, we need that you consider to rephrase the comment from reviewer 1 on page 6. 

Reviewer 1’s comment has been addressed in the manuscript. 

Additionally, after uploading figures to PACE, figures with individual subject lines (fig5-fig7) became blurred by the thinness and lightness of the individual subject traces. We have edited the lines so that they are darker and clearer after being uploaded to PACE. 

Reviewers' comments: 

Reviewer #1: The authors have addressed all of my comments with one exception. I feel that their characterization of the relation between the present study and Walter & Bex (2021) was a bit vague. On p. 6, they wrote that "In the present study, we utilize the same paradigm...". If I understand correctly, they used the same data. In other words, in the present study, the authors reanalyzed the data that was published in the previous study. The new analyses are sufficiently different from those of the previous study, so I'm not suggesting that this is a problem. My point is simply that it should be clear to the reader that the data are the same. 

Other than this one minor point, I think that the paper is suitable for publication. The authors did an excellent job responding to my concerns and the revised version will make an interesting contribution to the literature. 

It has been emphasized in the introduction of the present study (pg. 5) that the data used in the present analysis are that of the study from Walter & Bex (2021).

---

## [Editor Report · Decision Letter 2]

2 Nov 2022

Low-level factors increase gaze-guidance under cognitive load: a comparison of image-salience and semantic-salience models

PONE-D-22-06433R2

Dear Dr. Walter,

We’re pleased to inform you that your manuscript has been judged scientifically suitable for publication and will be formally accepted for publication once it meets all outstanding technical requirements.

Kind regards,

Marcela de Lourdes Peña Garay, Ph.D

Academic Editor

PLOS ONE
---

## [Editor Report · Acceptance letter]

5 Nov 2022

PONE-D-22-06433R2 

Low-level factors increase gaze-guidance under cognitive load: a comparison of image-salience and semantic-salience models 

Dear Dr. Walter:

I'm pleased to inform you that your manuscript has been deemed suitable for publication in PLOS ONE. Congratulations! Your manuscript is now with our production department. 

Kind regards, 

on behalf of

Dr. Marcela de Lourdes Peña Garay 

Academic Editor

PLOS ONE